# Nurse and Nursing Students’ Opinions and Perceptions of Enteral Nutrition by Nasogastric Tube in Palliative Care

**DOI:** 10.3390/nu13020402

**Published:** 2021-01-27

**Authors:** Eduardo Sánchez-Sánchez, Guillermo Ramírez-Vargas, Alicia Peinado-Canas, Francisco Martín-Estrada, Jara Díaz-Jimenez, Francisco Javier Ordonez

**Affiliations:** 1Punta de Europa Hospital, Algeciras, 11207 Cádiz, Spain; guiram1992@gmail.com (G.R.-V.); alipeinadoc@gmail.com (A.P.-C.); franciscomt400@gmail.com (F.M.-E.); 2Faculty of Education Sciences, University of Cádiz, 11519 Puerto Real, Spain; luna_nueva17@hotmail.com; 3Human Anatomy, School of Medicine, University of Cádiz, Plaza Fragela s/n, 11003 Cadiz, Spain; franciscojavier.ordonez@uca.es

**Keywords:** artificial nutrition, enteral nutrition, nasogastric tube, palliative care, tube feeding

## Abstract

It is widely accepted that nursing staff play a key role in palliative care (PC). The use of Nasogastric tubes (NG tubes) for Enteral Nutrition (EN) administration is still controversial in patients who receive PC. The aim of this study was to describe nurses’ and nursing students´ opinions and perceptions about EN using NG tubes in adult patients in palliative care. To achieve this goal, a cross-sectional descriptive study was carried out. A self-administered, semi-structured questionnaire intended for nurses and nursing students was used. Data was descriptively and inferentially analyzed using a chi-square test to determine the differential frequency of responses. In addition, a multivariate logistic regression model was also conducted. A total of 511 participants completed the questionnaire. Among them, nursing staff represented 74.9% (*n* = 383) whereas nursing students were 25.1% (*n* = 128). When life expectancy was above six months, 90.0% (*n* = 460) reported that EN using NG should be implemented. In contrast, when life expectancy is less than a month, 57.5% (*n* = 294) discouraged it. Significant differences within groups were found when life expectancy was <1 month (*p* = 0.044). It was also found that 491 participants (96.1%) reported that patient´s autonomy must be carefully respected for deciding whether continuing EN by NG tube or not. Finally, it was concluded for both nurses and nursing students that life expectancy should be the mean reason for implementing and withdrawing EN by NG tube. Major differences were found regarding when it should be ceased, suggesting perceptions may change as nurses graduate and move into their professional roles.

## 1. Introduction

Palliative Care (PC) is considered a model to follow in severely ill patients when curative treatment has been dismissed. Initially, the focus was on terminal oncology patients; however, over the years, other pathologies have been added, such as advanced dementia, respiratory diseases, etc. [1]. The goal of PC is to encourage comfort and to reach an optimum quality of life level [2]. 

Nutrition and hydration are both basic elements for life maintenance and are devised as signs of health in our society [3]. In general terms, patients in PC may have difficulties in keeping adequate ingestion due to lack of appetite, which may affect not only quality of life but also their relationships with relatives and health professionals [4]. Consequently, Artificial Nutrition and Hydration (ANH) has been widely implemented in clinical settings for both elderly [5] and young people [6] entering the end-of-life phase. 

However, the use of nasogastric (NG) tubes to administrate enteral nutrition (EN) is still being controversial, especially in patients near the end of their life [7]. Previous studies have reported some patients that do not wish to have a nasogastric tube placed [8], and this can be difficult to understand for relatives and health professionals [2]. Last but not least, it may lead to ethical dilemmas for healthcare professionals when planning nutritional interventions in PC [9,10]. 

In this respect, it is widely accepted that nursing staff may play a key role in PC in order to promote a consensus between appropriate treatment choices and the patients’ preferences that may finally facilitate the decision of applying this process [11,12]. Accordingly, the lack of knowledge and skills in PC from nurses may lead to erroneous perceptions related to approaching these patients [13]. Previous studies have reported that this lack of knowledge or skills referring to end-of-life care started during the academic stage [14]. Although experience seems to be a factor that improves skills in the management of patients with PC, this is not always the case that this experience is not supported by an update of knowledge based on new evidence.

For the reasons above mentioned, the aim of this study is to determine nurses and nursing students’ opinions and perceptions about EN using NG tube in adult patients with PC, thus knowing if academic knowledge or experience influence these opinions and perceptions, and therefore if they affect the approach to these patients.

## 2. Materials and Methods

### 2.1. Selection of Participants and Study Design. 

A cross-sectional descriptive study was used based on a self-administered and anonymous questionnaire intended for nurses and student nurses of any year (from first to fourth year). This was a non-probabilistic sample properly used as the questionnaire was sent to all nurses and student nurses from the whole Spanish territory. 

It was taken as reference for the calculation of the sample size the data published in the last Health National System report in 2017, and it reflected that a total of 245,533 nurses perform healthcare activities in Spain [15]. Data about the number of students in Nursing Degree programs were not available, but according to the Statistical Yearbook of Spain of 2019, the number of students in the field of Healthcare Sciences was 242,376, during the academic year of 2017–2018 [16]; consequently, this value was taken as a reference. The calculation of the sample size was carried out with a 95% confidence level and a precision of 3%; since the expected proportion of the change in population was unknown, a 0.5 proportion was selected. The sample size was 384 subjects.

### 2.2. Instruments and Variables

No standardized questionnaire exists for the context of the research that values the perceptions of nurses and/or student nurses about AN and the use of EN through NG tube in patients that receive PC. The questionnaire validated by Albanesi et al. [17] only included nurses working in oncology or palliative care (hospice or home care) who had experience in caring for patients with cancer and ANH.

Therefore, the researchers designed a semi-structured questionnaire based on doubts and opinions of colleagues along with their experience, together with topics of discussion presented in nutritional and palliative care training and nutrition conventions. A pilot study was carried out among colleagues from our ward and university students, from different academic years and during their practical period at hospitals, to verify the effectiveness of the questionnaire. That provided the necessary information to decide if we needed to modify any of the questions. After this pilot study, some questions were modified, due to the fact that some of the questions were not understood, something that could lead to an interpretation error. 

The first part of the questionnaire included sociodemographic variables, such as age (age range) and gender; and other variables, such as current workplace, with special attention to those units that had greater access to patients that receive palliative care (Primary health care, socio-medical centers, palliative care, geriatrics, internal medicine, oncology, and hematology) and whether they had worked or had carried out clinical practices with patients that had received palliative care. The second part included questions about initial opinions, maintenance, and cessation of EN by NG tube in adult patients receiving PC.

### 2.3. Data Collection

Concerning the distribution scheme of the questionnaire, we used new communication technologies such as the Google questionnaire platform, and to convey it, we used the social platforms Twitter, Facebook, WhatsApp, and Instagram. The questionnaire was distributed during June 2019.

### 2.4. Statistical Analysis

The data obtained from the variables were represented in a descriptive way. The qualitative variables were represented by frequency and percentages, and the quantitative variables were expressed by means of standard deviation or dispersion. Subsequently, by the use of a X^2^ test, significant differences between different groups were studied, taking as reference the nurses’ group, students were examined in their first to fourth years of a nursing university degree, and the difference in answers was tested, accepting a confidence level of 95%. Furthermore, a multivariate logistic regression model was carried out to study which factors of the first part of the questionnaire were significant in the answers about the initiation of EN by NG tube in different patients taking life expectancy as a reference. The statistical analysis was carried out using the R-Commander program. 

## 3. Results

### 3.1. Participant Characteristics.

We obtained a total of 511 answers to the questionnaire. From the total of the sample, 84.3% (*n* = 431) were females and 15.7% (*n* = 80) were males. According to age range, the 26–40-year-old range was the most predominant at 33.5% (*n* = 171), and the > 56-year-old interval had the lowest percentage (*n* = 18; 3.5%). 

Nurses represented 74.9% (*n* = 383) of the participants whereas nursing students were 25.1% (*n* = 128). In more detail, 11.0% (*n* = 56) were first- and second-year students and 14.1% (*n* = 72) were third- and fourth-year nursing students. Within different units of work, there were 72 nurses that practiced their role in Primary healthcare (14.1%); on the other hand, 3.7% (*n* = 19) of nurses worked in palliative care units. The other categories included units such as dialysis, traumatology, surgery, and cardiology. Some students responded that they worked in different wards; this is because they were, at the same time, health professionals of other categories while studying their nursing degree. 

Regarding the question related to work, having worked or done clinical practice with patients that receive PC, 76.8% (*n* = 408) answered affirmatively, and 20.2% (*n* = 103) answered that they had not. In all of these variables, there were some significant differences between the values from a statistical point of view (*p* < 0.001) (Table 1).

### 3.2. EN by NG Tube in Palliative Care: Knowledge and Opinions.

In Table 2, the answers to the questions of the questionnaire of different groups can be observed. Of the surveyed subjects, 65.9% (*n* = 337) responded that nutrition is important in palliative care, 30.1% (*n* = 154) said that it was important depending on their health condition, and the rest (*n* = 20; 3.9%) answered that nutrition has a secondary role in patients that receive palliative care. Statistically, significant differences appeared depending on the group (*p* = 0.004). This group of professional nurses considered that nutrition was important in palliative care, according to the health condition of the patients. 

When asking about which figure or figures are demanding the beginning or monitoring of ANH in PC, the most prevalent answer refers to patients, relatives, careers, and nurses with 57.7% (*n* = 295), and the highest value was shown by the group of student nurses in their first and second years (*n* = 47; 83.9%). When asked which aspect is the most important that must be assessed before starting ANH in patients that receive palliative care, the autonomy of the patient represented 19.5% (*n* = 100), symptom control (*n* = 109, 21.3%), life expectancy (*n* = 110; 21.5%), and nutritional condition (*n* = 175; 34.3%) received the largest number of answers. Statically significant differences within the three groups were observed (*p* = 0.008).

Regarding life expectancy and the initiation of EN by NG tube, answer percentages changed depending if life expectancy was >6 months, from 1 to 6 months, or <1 month. When life expectancy was above 6 months 90.0% (*n* = 460) believed that it should be initiated; also when it was 1 to 6 months, 81.6% thought that it is adequate to provide EN by NG tube. However, when life expectancy was less than a month, 57.5% (*n* = 294) disagreement started to show. There were significant differences within groups only when life expectancy was <1 month (*p* = 0.044). In this case, 60.6% (*n* = 232) of the group of nurses stated that EN by NG tube should be initiated, opposite to 51.8% (*n* = 29) of the group of students from first and second year and the 54.2% (*n* = 39) of the group of students in their third and fourth years. 

Of the studied sample, 58.9% (*n* = 301) responded that not feeding using EN by NG tube is not negative for a patient at the end of their lives. Significant differences were observed between groups (*p* < 0.001). The percentage of students that answered that it was negative, leaving a patient on end of life without feeding by NG tube, was 58.9% (*n* = 33) in the first and second year students as opposed to 40.3% (*n* = 29) in third and fourth year students, and 22.7% (*n* = 87) of the nurse group. Moreover, 71.0% (*n* = 363) of the subjects surveyed related that EN by NG tube must be suspended in patients in the last days of life, being the most prevalent answer within the nurse group (*n* = 287; 74.9%) and less in the first and second year student group (*n* = 31; 55.4%) (*p* = 0.002).

Concerning the question of whether the autonomy of the patient must be respected in the decision of continuing or not EN by NG tube, 491 (96.1%) of the surveyed responded affirmatively, not presenting any statistically significant differences between the different groups (*p* = 0.068).

A multivariate logistic regression model was carried out for the question related to the beginning of EN by NG tube and life expectancy, obtaining odds ratio (OR) values and the statistical significance using a 95% confidence interval. The outcomes obtained are shown in Table 3. In the table, it can be observed that all the studied variables did not have statistical significance when life expectancy > 6 months was taken as reference. If life expectancy was between 1–6 months, the subjects that worked in some specific units of work had a higher probability of responding affirmatively to the question about the initiation of EN by NG tube. If life expectancy was <1 month, the students and nurses that worked in different units showed a probability answer (OR 2.30 and OR 2.04, respectively) twice as high that they would start EN in these patients. The variable ‘woman’ had a statistical significance, but the OR value was less than 1. 

The rest of the variables have presented statistical significance for any of the previous conditions (Table 3).

## 4. Discussion

The results obtained show that 65.9% of the subjects considered that nutrition is important in the management of patients receiving PC and that nutritional status should be taken into account before starting ANH.

Nine out of ten of the participants answered that EN should be started with NG tube in patients with a life expectancy >6 months, but this percentage decreased when life expectancy was lower, reaching 42.5% in patients with a life expectancy <1 month, this response being more prevalent in nurses than in students.

A majority of subjects (58.9%) reported that not feeding EN due to NG tube in patients in the last days phase is not negative for the patient, this percentage was higher for nurses (63.4%), perhaps because the group of nurses considered EN with NG tube should be discontinued in patients in the last days phase.

Feeding and nutrition play an important role and, at the same time, are controversial issues within the treatment of palliative care patients. Sometimes, all this has made health professionals confirm that nutrition does not have an important role in these patients [18]. However, the subjects of our study confirm that nutrition does have an important role.

That is why, usage of AN may be justified in patients that receive palliative care, are undernourished, and have a reasonable life expectancy, if it would improve their quality of life [19,20]; but there are not enough good quality studies that help to provide recommendations about EN by nasogastric tube in patients that receive palliative care [21]. Occasionally, it is considered an unnecessary treatment that only contributes to extend suffering, and that this treatment should not be used [22]. Although it is known that a decrease in nutritional intake may lead to reduced comfort in these patients and consequently their quality of life, it may be perceived negatively by the patients and/or relatives [23]. 

According to the reported data from scientific literature, one of the factors that may influence making the decision to implement or continue EN by NG tube is life expectancy [2]. Regarding the question drafted in our questionnaire, the questions were formulated bearing in mind that this was the most important factor, but it was somehow difficult to separate the answers, as everyone can be influenced when making vital decisions. Even with these limitations within the study, the most prevalent factor was nutritional condition. This data agrees with data obtained by Orreval et al. in 2013, as the most common indication for EN by NG tube initiation was chewing and/or swallowing difficulties, regardless of the expected survival [7]. In our study, life expectancy occupied the second most prevalent factor. 

The nurses and student nurses surveyed are in favor of using EN by NG tube in patients within palliative care with a life expectancy of 1 to 6 months. But the percentage of professionals in favor of EN by NG tube decreases if life expectancy is less than a month. This is in line with the revised bibliography, where different professionals are in favor of not starting EN at the same time that life expectancy decreases, opposite to other treatments like antibiotics or the use of intravenous fluids [24]. These decisions may create ethical dilemmas and are related to feelings, thoughts, or beliefs [25].

The above can be associated with the next question of our questionnaire, about whether not feeding patients in the last days of life using EN by NG tube is something negative. The decision to not feed by ANH might be misunderstood as ‘not feeding’, as nutrition is associated with life and its absence with hunger. Occasionally a feeding care plan must be developed and can be called ‘feeding for comfort’ [2]; in other words, that patients could have what they fancy when they feel like eating. The results of our study match with this affirmation because almost six out of ten thought that not giving nutrition by NG tube was not negative.

The withdrawal of enteral feeding by NG tube in patients in the last days of life is controversial because maybe the family and/or nurses consider withholding EN, but the patient may prefer to withdraw this ANH. In the current study, 71.0% of participants would suspend the enteral feeding by NG tube in patients on their last days. In the study carried out by Seol et al., 34.3% of the survey agreed, with the inclusion of EN in the list of interventions that may be rejected by patients in the last days of life [25].

Sometimes, the patients themselves do not wish to have or to stop EN by NG tube. [8]. The suspension of EN by NG tube may cause ethical and legal problems. For instance, in South Korea, the law stipulates that ANH cannot be withdrawn. In the US, this right of the patient is recognized, as the patient Nancy Cruzan rejected ANH, and there were already guidelines in clinical practice in regards to the withdrawal or suspension of ANH [25]. 

There were many survey respondents in our study that believed that the patient must be the one that should decide (96.1%); this value was higher than the reported by other studies that took doctors as a reference (74.1%) [26]. For this reason, although there are no contraindications for its use, an individualized decision must be taken into account, respecting the principles of autonomy, benefits, and possible harm [27]. So, health professional teams that provide care to patients must establish which are the goals that they want to reach and if they can be reached, benefits that can be provided, and potential damage that they would cause [28]. Therefore, the principle of autonomy recognizes the right and capacity of a person to make a personal decision. Self-determination includes the right to refuse EN, although this refusal can be difficult to understand by relatives and health professionals [2]. Perhaps the way to avoid ethical conflicts and future dilemmas is the use of advanced guidelines to capture the patient´s decision about treatment or future techniques. Although the prevalence of patients that use this mechanism is very low [29]. 

The large number of participants in the survey should be considered a major strength of the study. In addition, participants were from different colleges of nursing and clinical settings in the whole of Spain, providing a high geographical representation.

The last point shows that the current study also had several limitations that should be considered when interpreting the findings. First, a potential selection bias, as an error, may exist, showing that participants were willing to complete the questionnaire because they were more aware of the importance of EN by NT than other colleagues.

## 5. Conclusions

The conclusions are that life expectancy was the primary reason for starting and withdrawing the EN by NG tube for both nurses and nursing students. Major differences between nurses’ and nursing students´ perceptions were found regarding the importance of EN by NG tube in PC and the choice of when it should be ceased in patients in the last days of life. The latter results suggest that the perceptions changed as nurses graduate and move into their professional roles.

## Figures and Tables

**Table 1 nutrients-13-00402-t001:** Socio-demographics variables.

	Nurse	Student (1^st^ and 2^nd^)	Student (3^rd^ and 4^th^)	*p*-Value	Total
	*n* (%)	*n* (%)	*n* (%)		*n* (%)
**Gender**					0.934
Female	323 (84.3%)	48 (85.7%)	60 (83.3%)	431 (84.3%)	
Male	60 (15.7%)	8 (14.3%)	12 (16.7%)	80 (15.7%)	
**Age intervals**					<0.001**
≤25 years	53 (13.8%)	50 (89.3%)	58 (80.6%)	161 (31.5%)	
26–40 years	154 (40.2%)	5 (8.9%)	12 (16.7%)	171 (33.5%)	
41–55 years	158 (41.3%)	1 (1.8%)	2 (2.8%)	161 (31.5%)	
≥56 years	18 (4.7%)	0 (0.0%)	0 (0.0%)	18 (3.5%)	
**Unit or Centre of work**					<0.001**
Primary Healthcare	69 (18.0%)	0 (0.0%)	3 (4.2%)	72 (14.1%)	
Socio-sanitary Centre’s	20 (5.2%)	0 (0.0%)	0 (0.0%)	20 (3.9%)	
Palliative care	18 (4.7%)	0 (0.0%)	1 (1.4%)	19 (3.7%)	
Geriatrics	24 (6.3%)	0 (0.0%)	3 (4.2%)	27 (5.3%)	
Internal Medicine	41 (10.7%)	1 (1.8%)	1(1.4%)	43 (8.4%)	
Oncology-Hematology	31 (8.1%)	0 (0.0%)	1(1.4%)	32 (6.3%)	
Others	179 (46.7%)	2 (3.6%)	3 (4.2%)	114 (22.3%)	
Student	1 (0.3%)	53 (94.6%)	60 (83.3%)	184 (36.0%)	
**Work/has worked/has carried out practices with patients that receive palliative care:**					<0.001**
YES	336 (87.7%)	20 (35.7%)	52 (72.2%)	408 (79.8%)	
NO	47 (12.3%)	36 (64.3%)	20 (27.8%)	103 (20.2%)	

*p*-value: **0.001.

**Table 2 nutrients-13-00402-t002:** Distribution of responses to the questionnaire.

	Nurse	Student	Student	Total	
	*n* (%)	(1^st^ and 2^nd^)	(3^rd^ and 4^th^)	*n* (%)	*p*-Value
		*n* (%)	*n* (%)		
**Regarding nutrition, do you believe that is important in palliative care?**					0.004*
YES	236 (61.6%)	47 (83.9%)	54 (75.0%)	337 (65.9%)	
Depending on health condition	132 (34.5%)	7 (12.5%)	15 (20.8%)	154 (30.1%)	
Not important, plays a secondary role	15 (3.9%)	2 (3.6%)	3 (4.2%)	20 (3.9%)	
**Who demand the starting or maintenance of artificial nutrition and hydration in palliative care?**					<0.001**
Patients	10 (2.6%)	0 (0.0%)	1 (1.4%)	11 (2.1%)	
Relatives/carers	129 (33.7%)	2 (3.6%)	28 (38.9%)	159 (31.1%)	
Nurses	18 (4.7%)	4 (7.1%)	5 (6.9%)	27 (5.3%)	
All of them	211 (55.1%)	47 (83.9%)	37 (51.4%)	295 (57.7%)	
Not demanded by any of them	15 (3.9%)	3 (5.4%)	1 (1.4%)	19 (3.7%)	
**Which essential aspect should be valued before starting enteral feeding in these patients? Mark the most important one:**					0.008*
Autonomy of the patient	68(17.8%)	16 (28.6%)	16(22.2%)	100 (19.5%)	
Symptoms control	83(21.7%)	4 (7.1%)	22(30.6%)	109 (21.3%)	
Life expectancy	90(23.5%)	11(19.6%)	9 (12.5%)	110 (21.5%)	
Nutritional condition	126 (33.9%)	25(44.6%)	24 (33.3%)	175 (34.3%)	
Others	16 (4.2%)	0 (0.0%)	1 (1.4%)	17 (3.3%)	
**Would you initiate enteral feeding in patients with palliative care with a life expectancy > than 6 months?**					0.387
YES	345 (90.1%)	48 (85.7%)	67 (93.1%)	460 (90.0%)	
NO	38 (9.9%)	8 (14.3%)	5 (6.9%)	51 (10.0%)	
**Would you initiate enteral feeding in patients with palliative care with a life expectancy between 1 to 6 months?**					0.237
YES	307 (80.2%)	50 (89.3%)	60 (83.3%)	417 (81.6%)	
NO	76 (19.8%)	6 (10.7%)	12 (16.7%)	94 (18.4%)	
**Would you initiate enteral feeding in patients with palliative care with a life expectancy < than 1 month?**					0.044*
YES	232 (60.6%)	29 (51.8%)	39 (54.2%)	217 (42.5%)	
NO	151 (39.4%)	27 (48.2%)	33 (45.8%)	294 (57.5%)	
**Do you believe that not nourishing a patient in their last days phase, using enteral nutrition by tube, is something negative?**					<0.001**
YES	87(22.7%)	33 (59.0%)	29 (40.3%)	149 (29.2%)	
In these patients is not important to nourish	53(13.8%)	5 (8.9%)	3 (4.29%)	61 (11.9%)	
NO	243 (63.4%)	18 (32.1%)	40 (55.6%)	301 (58.9%)	
**In case of enteral feeding by NG tube, should be suspended in patients in last days phase?**					0.002*
YES	287 (74.9%)	31 (55.4%)	45 (62.5%)	363 (71.0%)	
NO	96 (25.1%)	25 (44.6%)	27 (37.5%)	148 (29.0%)	
**Patient’s autonomy should be respected in the decision of continuing or not with enteral feeding by tube?**					0.068
YES	372 (97.1%)	51 (91.9%)	68 (94.4%)	491 (96.1)	
NO	11 (2.9%)	5 (8.9%)	4 (5.6%)	20 (3.9)	

*p*-value: *0.05; **0.001.

**Table 3 nutrients-13-00402-t003:** Odds Ratio Values.

	LE > 6 Months			LE 1–6 Months			LE <1 Month		
	OR	CI 95%	*p*-Value	OR	CI 95%	*p*-Value	OR	CI 95%	*p*-Value
**Age interval**									
26–40 years	1.63	0.38–47.08	0.271	1.14	0.56–2.31	0.704	1.15	0.66–2.08	0.608
41–55 years	1.3	0.67–3.96	0.551	0.94	0.45–1.92	0.873	0.91	0.50–1.65	0.765
< 56 years	2.41	0.52–3.14	0.427	0.8	0.24–3.01	0.734	0.64	0.18–1.99	0.459
**Unit or Centre of work**									
Socio-sanitary centers	0.65	0.15–3.37	0.578	1.02	0.34–3.29	0.967	0.25	0.03–1.02	0.09
Palliative care	0.59	0.14–3.09	0.498	0.89	0.29–2.80	0.84	0.58	0.15–1.88	0.396
Geriatrics	0.87	0.21–4.45	0.861	0.85	0.32–2.32	0.748	1.16	0.42–3.05	0.752
Internal Medicine	2.15	0.47–15.21	0.361	3.9	1.32–14.41	0.022*	2.15	0.96–4.87	0.062
Oncology-Hematology	3.36	0.54–65.09	0.27	4.24	1.29–19.33	0.030*	1.68	0.68–4.11	0.252
Student	1.34	0.39–4.22	0.622	2.82	1.13–7.01	0.026*	2.3	1.07–5.03	0.033*
Others	1.13	0.41–2.84	0.79	2.65	1.34–5.23	0.004**	2.04	1.13–3.80	0.020*
**Gender** (female)	0.71	0.26–1.66	0.476	0.7	0.33–1.35	0.316	0.52	0.31–0.87	0.013*
**Work/have worked or have performed practices** (YES)									
	1.83	0.89–3.66	0.089	1.37	0.73–2.51	0.306	0.9	0.55–1.45	0.623

*p*-value: *0.05; **0.001, LE: Life Expectancy; OR: Odds Ratio; CI: Confidence interval.

## Data Availability

The data is kept in a database prepared by the research team.

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
