# Peer review of "Nurse and Nursing Students’ Opinions and Perceptions of Enteral Nutrition by Nasogastric Tube in Palliative Care"

_nutrients, 2021, doi:10.3390/nu13020402_

Round 1
Reviewer 1 Report
Dear authors
Thanks for the opportunity to read the manuscript. I think it almost a very important topic but i have suggested some major comments.
Article
Thank you for the opportunity to review this article. I think it is almost a very important topicbut. I have suggested some major comments.
Abstract:
Line14: What exactly is controversial in dealing with nasogastric tubes (NG tube) for enteral nutrition (EN)? -is the application from NG tube meant? -Or whether feeding is to be done?
Introduction:
The introduction is clearly and comprehensibly described.From my point of view, the problem refers to end of life care and not to the entire phase of palliative care.What exactly are the problems of the NG tube in the end of life phase?
Materials and Methods:
Selection of Participants and Study Design.How exactly were the nursesselected in relation to their experience with PC? Do nurses have a similar understanding of PC?What criteria were used to select settings such as dialysis, etc.?
Data Collection:
Approximately how many nurses/students received the questionnaire?Were there reminders?
Results:
Participant Characteristics.Are there any guidelines for dealing with NG tubein PC in the different settings? Table 1: What is meant by others? It has a large number there?What is meant by student 1(0.3%)?
Discussion:
Start the discussion with a short summary of the main findings of the study. You talk about questions from your questionnaire in the discussion. I cannot understand this because I do not know the questionnaire. Can the questionnaire be attached?To what extent is the administration of nutrition also related to other measures that are no longer done or are done differently at the end of life?(Such as medications?).
Conclusion:
What do their results now mean for practice, nursing education and research?
Author Response
Dear reviewer,
Firstly, we appreciate the time dedicated to our manuscript, as well as the explanations that you ask for that help us to know doubts that a future reader may have, if the manuscript gets published.
Secondly, we answer to the questions that you have made, with aim of resolving doubts raised by our manuscript.
Abstract:
Line14: What exactly is controversial in dealing with nasogastric tubes (NG tube) for enteral nutrition (EN)? -is the application from NG tube meant? -Or whether feeding is to be done?
Feeding itself is controversial in the last days phase, but in this case we talk about the use of nasogastric tube for EN as we refer to patients receiving palliative care being or not in their last days.
Introduction:
The introduction is clearly and comprehensibly described.From my point of view, the problem refers to end of life care and not to the entire phase of palliative care.What exactly are the problems of the NG tube in the end of life phase?
We understand that the problem starts when the patient and/or the family knows there is not a curative treatment for the disease, and that produces negative effects in the psychological state. This alteration, along with health problems, influence in the feeding of these patients negatively. That is the reason why, in some occasions NG tube for Enteral Feeding must be used for its administration, as this patients may have a life expectancy of months and a nutritional déficit may lead to comorbidities increase (inmune system alterations, injuries,…). Due to everything that involves these kind of patients may lead to problems and disputes with professionals and/or relatives, that is why nurses play an important role as a care guarantor and health education.
As you said, in last days phase, this problem is exacerbated, because EN by NG tube might be seen as something that do not contribute and make them feel uncomfortable. This idea can be also common within nurses attending these patients, either because they diminish the importance of nutrition and/or believe that procedures that prolong the life in these patients should not be carried out.
Materials and Methods:
Selection of Participants and Study Design.How exactly were the nursesselected in relation to their experience with PC? Do nurses have a similar understanding of PC?What criteria were used to select settings such as dialysis, etc.?
The questionnaire was disseminated through social networks and it was sent to nursing councils, health centres,… Nurses were not selected according to their experience in palliative care, but every nurse that voluntary wanted to participate, no matter which ward they worked. The reason why this kind of selection was carried out, is shown below:
- Currently, there is not an speciality in Nursing for Palliative Care in Spain, although there is a progress on nurses training in this field.
- Patients with palliative care are not always attended by nurses with advance knowledge in palliative care, as the patient might be assisted due to other health problems or in general wards instead of specialised palliative care units.
- There are rotation and mobility within different wards of centres, due to types of contract, internal mobility, relocations,…
In enviroments like dialysis or other units, palliative care patients might be attended and their assistance activity influence on nutritional approach of these patients.
Data Collection:
Approximately how many nurses/students received the questionnaire?Were there reminders?
With the aim of reaching the whole national territory, the questionnaire was sent through an online link and it was disseminated using social networks, nursing council, healh centres,… We do not know the number of nurses and student nurses that received it, only the number of those who answered it.
Reminders were not made, as the objective of this study was to determine opinions and perceptions of nurses and nursing students about EN by NG tubes in patients with PC, to visualize any problems that may exist, because it may influence in the assistance that it is provided to these patients. Results help us to suggest improvements like a clinical practical guideline in the approach of EN in these patients, especially that one by NG tube.
Results:
Participant Characteristics.Are there any guidelines for dealing with NG tubein PC in the different settings? Table 1: What is meant by others? It has a large number there?What is meant by student 1(0.3%)?
There are patterns, guidelines or recommendations about the use of EN by nasogastric tube in different kind of patients or procedures, this is more than in specific units. So, the nutritional approach of a palliative care patient should be the same one than in other units and value the patients as well as their health condition.
Others. It is difficult to encompass all the units where nurses perform their assistance activities, because there are many medical specialties. The term ‘OTHERS’ encompass units like: Dialysis, Traumatology, Surgery and Cardiology… puta ll together as these wards may have sporadic contact with this kind of patients.
The figure of 1 (0.3%) in table 1, mean that one of the nurses did not practice assistance activity and she continued with her trainings, so she is considered registered nurse, student, but not active.
Discussion:
Start the discussion with a short summary of the main findings of the study. You talk about questions from your questionnaire in the discussion. I cannot understand this because I do not know the questionnaire. Can the questionnaire be attached?To what extent is the administration of nutrition also related to other measures that are no longer done or are done differently at the end of life?(Such as medications?).
As you advised us, discussion has started with a brief summary about the main findings of the study. The modified manuscript has been attached.
The questionnaire has not been attached because the authors have decided that would be a better explanation and better to visualise the ítems in the tables. That is the reason why in the first column of table 1 and 2 appear all the questuons of the questionnaire. Although if you believe it is necessary an it helps to a better understanding of the study, we can attach it.
Pengo et sl in 2017 carried out a qualitative study where they wanted to know nurses and doctors opinions about the management of different measures including the use of artificial nutrition in patients with advanced dementia and patients with different life expectancies from 1 to 6 months. The percentage of surveyed that agreed with the use of artificial nutrition was 71%, but if life expectancy was less than a month, this percentage decreased until 48%. This decrease was more acute in comparison to other measures, for instance, antibiotics administration (83% against 73%) or artificial hydration (79% against 61%)*.
Medication management is different to the nutrition one, because it is erroneusly believed that medicines improve patient’s health but nutrition is something secondary.
*Pengo V, Zurlo A, Voci A, et al. Advanced dementia: opinions of physicians and nurses about antibiotic therapy, artificial hydration and nutrition in patients with different life expectancies. Geriatr Gerontol Int. 2017;17(3):487-493. doi:10.1111/ggi.12746
Conclusion:
What do their results now mean for practice, nursing education and research?
Results shown that exist a variability between nursing students and nurses, and also between nurses depending on their ward where they work. This make us think the need of créate a clinical practice guideline about nutritional approach of these patients, where the use of NG tubes are included. It is a must to keep moving forward in the education so these differences do not exist and clinical practice become the same for all nurses.
Moreover, this visualize the need to promote the Advance Vital Directives Document to have a document that reflect patients choice, when cannot participate in their decision making.
Investigations should be made about benefits and risks of the use of EN by NG tube in this patients, so care provided are based on evidence. Clear evidence would help to reduce the variability in the management of these patients.
Once again, we appreciate the time and attention dedicated to our manuscript. We really hope we have reached your expectations, with the modifications made and that the explanations to those that we have not modified be considered as appropiate.
Kind regards.

Reviewer 2 Report
this is a survey to nurses and nursing students in Spain about use of nasoenteric feeding near end of life in patients receiving palliative care. It is curious that the focus of this survey is only on nurses give that palliative care is an interdisciplinary activity. It would have been valuable to have patients or other HCP included. The distinctions noted between nurses and nursing students at various levels of training is what is expected as individuals mature into their profession and get more experience - this is not really clearly described nor is literature cited. The context of country is important - can we generalize these findings beyond Spain? - what are the implications?
From a methods perspective the authors have achieved a good response rate to the survey - triangulation of the findings with either focus groups or interviews to more fullsomely explore the information would be of interest.
Author Response
Dear reviewer,
Firstly, we appreciate the time dedicated to our manuscript, as well as the explanations that you ask for that help us to know doubts that a future reader may have, if the manuscript gets published.
Secondly, we answer to the questions that you have made, with aim of resolving doubts raised by our manuscript.
This is a survey to nurses and nursing students in Spain about use of nasoenteric feeding near end of life in patients receiving palliative care. It is curious that the focus of this survey is only on nurses give that palliative care is an interdisciplinary activity. It would have been valuable to have patients or other HCP included.
As you mention, palliative care is an interdisciplinary activity, however we have carried out the study woth these professionals because they have a close contact with these kind of patients, either ambulatory or hospital level. Nurses are the health agents that help with patients self-care, and nutrition is one of them.
These results have helped us to improve domiciliary assistance in these patients from Palliative Care Domiciliary Support Team and to initiate interviewing patients and relatives with the aim of reaching common points and reflect it in a clínical practice guideline for nutrition approach in these patients.
The distinctions noted between nurses and nursing students at various levels of training is what is expected as individuals mature into their profession and get more experience - this is not really clearly described nor is literature cited.
As you said, differences may be caused by the experience and maturity in their assitance activity, but also this experience may appeared to be against us. Sometimes, experience makes obsolete and without evidence knowledge not to get updated, that is why we made a comparison between nurses and nursing students. Added in the introduction.
The context of country is important - can we generalize these findings beyond Spain?
Data might be extrapolated to other countries, where health assistance seems like our. In other countries similar results have been obtained in the study, so the creation of a clinical practice guideline may be extrapolated to other countries. We must not forget, that the patient would have same caracteristics and the management would be very similar.
What are the implications?
Results shown that exist a variability between nursing students and nurses, and also between nurses depending on their ward where they work. This make us think the need of créate a clinical practice guideline about nutritional approach of these patients, where the use of NG tubes are included. It is a must to keep moving forward in the education so these differences do not exist and clinical practice become the same for all nurses.
Moreover, this visualize the need to promote the Advance Vital Directives Document to have a document that reflect patients choice, when cannot participate in their decision making.
Investigations should be made about benefits and risks of the use of EN by NG tube in this patients, so care provided are based on evidence. Clear evidence would help to reduce the variability in the management of these patients.
From a methods perspective the authors have achieved a good response rate to the survey - triangulation of the findings with either focus groups or interviews to more fullsomely explore the information would be of interest.
Meetings with focus groups were made to know if the questionnaire was appropiate as it was created by authors. The triangulation of the finding it is something interesting and I have suggested to the other authors to carry it out, as well as interviews, but the situation at the moment makes it difficult for us to arrange these interviews directly and even by telephone.
Once again, we appreciate the time and attention dedicated to our manuscript. We really hope we have reached your expectations, with the modifications made and that the explanations to those that we have not modified be considered as appropiate.
Kind regards.
Round 2
Reviewer 1 Report
Dear authors
Thank you for resubmitting the manuscript and these careful revisions. I do not have any comments furthermore.
Author Response
Estimado revisor,
Nos complace saber que hemos respondido satisfactoriamente a sus propuestas.
La traducción al inglés ha sido revisada por colegas del Reino Unido y un colega que ha realizado traducciones oficiales. Todos ellos aparecen en los agradecimientos.
Saludos cordiales.
Reviewer 2 Report
thank you for the changes you have made. Although the quality of the science is not high the findings are somewhat novel and articulate an important point.
Author Response
Dear reviewer,
We are pleased to know that we have satisfactorily responded to your proposals.
English translation has been reviewed by colleagues from the UK and a colleague who has done official translations. All of them appear in the acknowledgments.
Kind regards.